# Estimating Soil Water Susceptibility to Salinization in the Mekong River Delta Using a Modified DRASTIC Model

**Thanh N. Le [1], Duy X. Tran [2], Thuong V. Tran [3],\*, Sangay Gyeltshen [4], Tan V. Lam [5,6], Tung H. Luu [1], Dung Q. Nguyen [1] and Tuyet V. Dao [7]**

1   Ho Chi Minh City Institute of Resources Geography, Vietnam Academy of Science and Technology, Ho Chi Minh City 70000, Vietnam; lnthanh@hcmig.vast.vn (T.N.L.); lhtung@hcmig.vast.vn (T.H.L.); nqdung@hcmig.vast.vn (D.Q.N.)
2   School of Agriculture and Environment, College of Sciences, Massey University, Palmerston North 4442, New Zealand; D.Tran@massey.ac.nz
3   Institute of Engineering and Technology, Thu Dau Mot University, Thu Dau Mot City 75000, Vietnam
4   Department of Remote Sensing and Geographic Information System, APECS Consultancy Services, Thimphu 11001, Bhutan; sangye89@gmail.com
5   Institute of Environmental Sciences, Nguyen Tat Thanh University, Ho Chi Minh City 70000, Vietnam; lvtan@ntt.edu.vn
6   Department of Science and Technology, People's Committee in Ben Tre, Ben Tre City 86000, Vietnam
7   Department of Management of Scientific Research and External Relations, Binh Duong University, Thu Dau Mot City 75000, Vietnam; dvtuyet@bdu.edu.vn
\*   Correspondence: thuong.tran@tdmu.edu.vn

**Abstract:** Saltwater intrusion risk assessment is a foundational step for preventing and controlling salinization in coastal regions. The Vietnamese Mekong Delta (VMD) is highly affected by drought and salinization threats, especially severe under the impacts of global climate change and the rapid development of an upstream hydropower dam system. This study aimed to apply a modified DRASTIC model, which combines the generic DRASTIC model with hydrological and anthropogenic factors (i.e., river catchment and land use), to examine seawater intrusion vulnerability in the soil-water-bearing layer in the Ben Tre province, located in the VMD. One hundred and fifty hand-auger samples for total dissolved solids (TDS) measurements, one of the reflected salinity parameters, were used to validate the results obtained with both the DRASTIC and modified DRASTIC models. The spatial analysis tools in the ArcGIS software (i.e., Kriging and data classification tools) were used to interpolate, classify, and map the input factors and salinization susceptibility in the study area. The results show that the vulnerability index values obtained from the DRASTIC and modified DRASTIC models were 36–128 and 55–163, respectively. The vulnerable indices increased from inland districts to coastal areas. The Ba Tri and Binh Dai districts were recorded as having very high vulnerability to salinization, while the Chau Thanh and Cho Lach districts were at a low vulnerability level. From the comparative analysis of the two models, it is obvious that the modified DRASTIC model with the inclusion of a river or canal network and agricultural practices factors enables better performance than the generic DRASTIC model. This enhancement is explained by the significant impact of anthropogenic activities on the salinization of soil water content. This study's results can be used as scientific implications for planners and decision-makers in river catchment and land-use management practices.

**Keywords:** salinity; bearing layer; agricultural practices; total dissolved solids (TDS); Ben Tre

## 1. Introduction

Groundwater, acknowledged as a critical part of the hydrological process, is an irreplaceable resource for anthropological activities [1] and soil humidity preservation in the dry season [2]. The coastal soil-water-bearing layer includes the active groundwater sources in the coastal areas that allow plants and other organisms to live [3,4]. Although fresh water

in the soil-water-bearing layer is often not adequate for human consumption, it plays an essential role in sustaining agricultural and natural environments [4,5]. Saltwater intrusion into the freshwater surface layer is a natural process that takes place in coastal regions due to changes in the hydrological systems and fluctuations in the groundwater table level [6,7]. Concurrent with drought, it severely damages soil layers around coastal regions [8,9]. This phenomenon has become a common issue in deltas and coastal lowlands, which are characterized by shallow water tables, insufficient drainage networks, and seawater intrusion [10]. Salinization is affected by both natural factors (e.g., the topography, climatic conditions, river/canal network, and hydrological regime) and anthropologic drivers (e.g., increased saline/brackish water demand for aquaculture) [5,11]. Hence, a consideration of the soil water vulnerability in coastal areas needs to take into account the effects of saltwater intrusion in conjunction with local socioeconomic and physical conditions.

Mapping groundwater vulnerability to salinization in coastal regions provides necessary information for the appropriate management of of groundwater resources [12,13]. Different approaches have been applied to estimate groundwater vulnerability, and can be grouped into three categories (i.e., index-based, statistical, and simulation-based approaches) [14–16]. With its simple and straightforward method, the DRASTIC model, which was primarily developed by the National Water Well Association and the U.S. Environmental Protection Agency, is among the most widely used approaches to groundwater vulnerability estimation [14,17]. However, due to the variation in local conditions (e.g., geological or geo-hydrological settings), removing factors, altering input factors, or adding new parameters such as land use or irrigation types may be conducted with the generic DRASTIC model to achieve a better outcome [18]. For example, Chenini et al. (2015) [19] removed two factors (i.e., Aquifer (A) and Hydraulic Conductivity (C)) to quantify the vulnerability index in the saturated aquifer zone of Grombalia. Singh et al. (2015) [20] added anthropic factors to the DRASTIC index to determine pollutants in an urban area in India. The results of both adjusted models were better than those of the classic model when they were validated using field data. Furthermore, validation has often been implemented because the DRASTIC model is an indirect method that mainly relies on accessible generic data [21,22]. A DRASTIC model that is not validated can result in an incorrect vulnerability assessment [12,23]. Therefore, this study applied an adjusted DRASTIC model to characterize the water salinity risk in the soil layer. Validation was also conducted to improve the accuracy of the model.

The Mekong River Delta (MRD), the most important region for rice production in Vietnam, is severely affected by the global climate change threats [9,24]. The impact is more severe due to the combined effects of upstream water reduction [25]. It is projected that, with the long-term decline in rainfall and upstream water and increased water consumption during the dry season, the salinization process will be more widespread, and therefore, will significantly threaten the water quality in the region [26,27]. Hence, assessing soil water vulnerability to salinization is an essential step in understanding the situation and process of water supply in coastal areas and developing better strategies for groundwater resources management. This achievement is valuable for mitigating land degradation and promoting sustainable agriculture development in the region [28]. Several studies examining the soil salinity patterns in the MRD revealed that infiltration of saline content into groundwater depends on the land use and river density [8,9,29]. Each land use type will have a different influence on water contamination. For instance, saltwater enters the bottom soil layer more quickly on agricultural land than other land use land cover (LULC) types by the direct use of water for agricultural production from a salted river or canal system in the dry season. Besides, saltwater intrusion will occur more easily in the regions that have a populated river density. Hence, modifying the classic model by combining the river catchment and land use as additional factors is critical to estimating the saltwater intrusion risk in the coastal soil-water-bearing layer in the MRD.

The primary purpose of this study was to model zones vulnerable to saltwater intrusion in the coastal area of MRD using a GIS-based Mod-DRASTIC model. The specific

objectives were to (i) quantify the spatial pattern of thematic factors for the Mod-DRASTIC model, (ii) assess the soil water susceptibility to salinization, and (iii) validate the accuracy of the model outcomes. The Ben Tre province, a coastal province located in the MRD, was selected as a case study to implement the research. The findings of this study will support the local government in proposing action strategies for preventing salinization in vulnerable areas and properly managing coastal groundwater resources.

## 2. Materials and Methods

### 2.1. Study Area

Ben Tre, one of the provinces in the Vietnamese MRD with an area of 2394.81 km$^2$, lies between 9°48′0′′ and 10°20′0′′ N latitude and 105°57′0′′ and 106°48′0′′ E longitude (Figure 1). The province borders tributaries of rivers and has four main estuaries (i.e., Dai, Ba Lai, Ham Luong, and Co Chien). It is located in an equatorial monsoon climate with an annual precipitation of 2000–2300 mm and an average annual temperature of 26–37 °C. Moreover, the province has a flat terrain with a topography below 1.0 m that is located around riparian and low-lying coastal regions, regularly inundated during high tide. Hence, it is quickly affected by mixing river and ocean dynamics throughout the year, and the influence of saltwater intrusion is aligned with drought through river mouths [30]. The impacts of salinization have become severe since 2003. The most severe salinity, with a concentration of 4‰, was discovered around major rivers and transmitted to 45–70 km from the coastal line during the dry season of 2015–2016. Therefore, the quality of groundwater in the surface horizon may be threatened by factors leading to salinization.

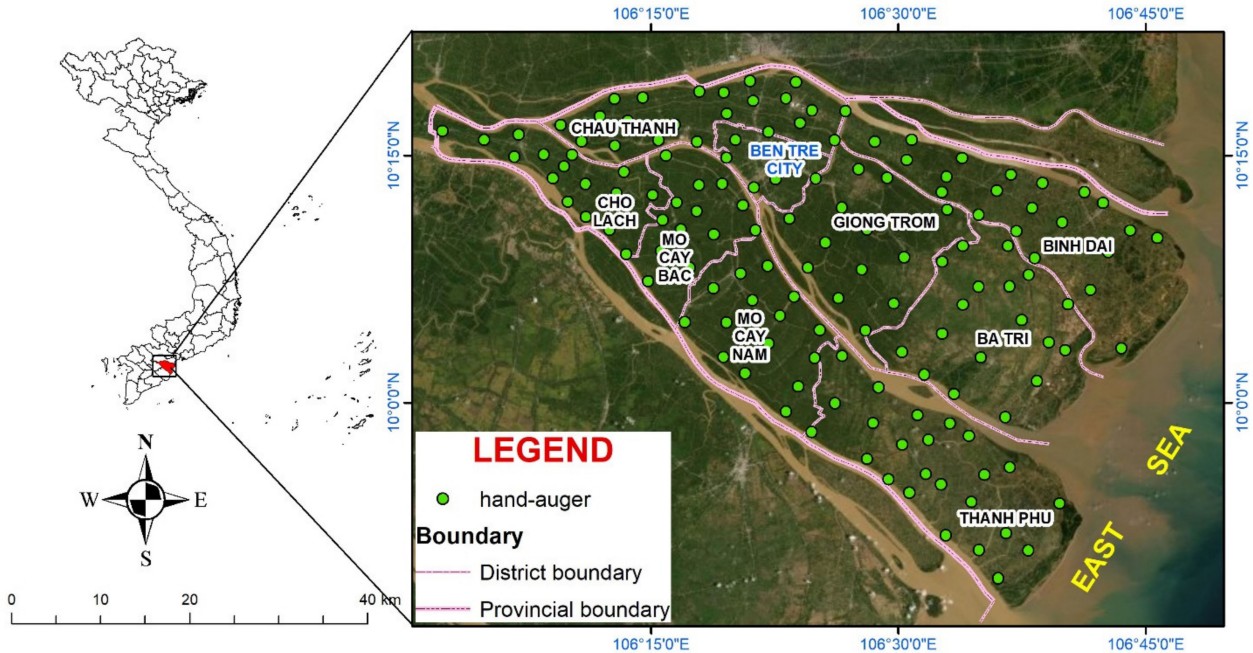

**Figure 1.** The study area and location of 150 hand-auger samples.

### 2.2. Data Collection

One hundred and fifty hand-auger tests with a depth of 100 cm and a spatial grid of 5 km were conducted in April 2018 to collect field data in the study site. The coordinate system of the pits was determined by using a Garmin (USA) handheld GPS-map 60CSx with an accuracy of ±5 m (Figure 1). A hand-auger location was selected to represent whole types of land use within the province. The 150 soil samples were taken at a weight of 2 kg (depth of 0–30 cm), while the soil water samples were collected with a volume of 2 L (depth of 10–100 cm). The soil and soil water samples were stored in a closed foam container with a constant temperature of 4 °C and were analyzed for total dissolved solids

(TDS) and electrical conductivity (EC, mS/cm) at the Center of Analysis, Experiment, and Mineral Technology, Ho Chi Minh City Institute of Resources Geography following the USDA system for soil samples and soil texture. The use of data with a scale of 1:50,000 to prepare the input factors is presented in Table 1.

**Table 1.** Data used for mapping the parameters of the model.

| Factor | Format | Sources |
|---|---|---|
| Depth of water (D) | point | Field data |
| Aquifer media (A) | point | Derived from field data |
| Net recharge (R) | polygon | |
| Soil media (S) | polygon | |
| Topography (T) | polygon | Department of Natural |
| Impact of vadose zone (I) | polygon | Resources and Environment |
| Hydraulic conductivity (C) | polygon | in Ben Tre province |
| River catchment (RC) | polygon | |
| Land use types (LU) | polygon | |

### 2.3. The Modified DRASTIC Model

Previous studies considered that the generic DRASTIC model used seven primary factors that closely related to geological and hydrogeological conditions to calculate the vulnerability index (i.e., depth of watertable, aquifer media, net recharge, soil media, topography, impact of vadose zone, and hydraulic conductivity) [14,17,31]. These factors play a substantial role in characterizing the presence and movement of groundwater in the aquifer system [32]. Furthermore, the vigor including/excluding the factors is a principal advantage of the DRASTIC approach in the context of typical features at each region and data availability [18]. Several studies developed a model similar to the modified DRASTIC index by adding the flexibility of lineament density or even land-use pattern [14,32–34]. This enables the characterization of the intrinsic properties of aquifers (i.e., static factors) by combining the land use factor with the generic DRASTIC model to assist with analyzing the anthropogenic influences (i.e., dynamic factors) [32].

In this study, a vulnerability to salinization index was quantified by using the nine thematic factors, applying prescribed weights of 1 to 5 and factor classes with prescribed ratings of 1 to 10, as in Equation (1):

$$DI = \sum_{i=1}^{n} W_i R_i \qquad (1)$$

where $DI$ is the Mod-DRASTIC index; $W_i$ and $R_i$ are the weight and rating of factor $i$, respectively. Nine parameters (Table 1), developed in the form of nine thematic map layers, were used as input for for the proposed Mod-DRASTIC model. The soil-water-bearing layer's vulnerability maps and thematic maps were prepared using ArcGIS v. 10.8 ([ESRI], Redlands, CA, USA) software.

### 2.4. Thematic Layer Processing

The DRASTIC model is flexible for modification as it allows users to define the model input and weight the importance of the model variables based on their experience and knowledge [23]. This ensures that local hydrogeological characteristics are taken into account, and therefore, the model suits better for the local case study [35,36]. In Equation (1), all the essential parameters are determined according to their key roles in the presence of soil water vulnerability in the study area. The weighting values of the modified DRASTIC are shown in Table 2. Finally, model calibration was conducted, as shown in Figure 2. Vulnerable areas of the soil-water-bearing layer were identified based on the DRASTIC index values (e.g., higher values indicate greater vulnerability to salinization).

**Table 2.** The specific weight of the factors in the Mod-DRASTIC model [32,33,37].

| Factor | Weight |
|---|---|
| Depth of water (D) | 5 |
| Aquifer media (A) | 4 |
| Net recharge (R) | 3 |
| Soil media (S) | 2 |
| Topography (T) | 2 |
| Impact of vadose zone (I) | 5 |
| Hydraulic conductivity (C) | 4 |
| River catchment (RC) | 5 |
| Land use pattern (LU) | 5 |

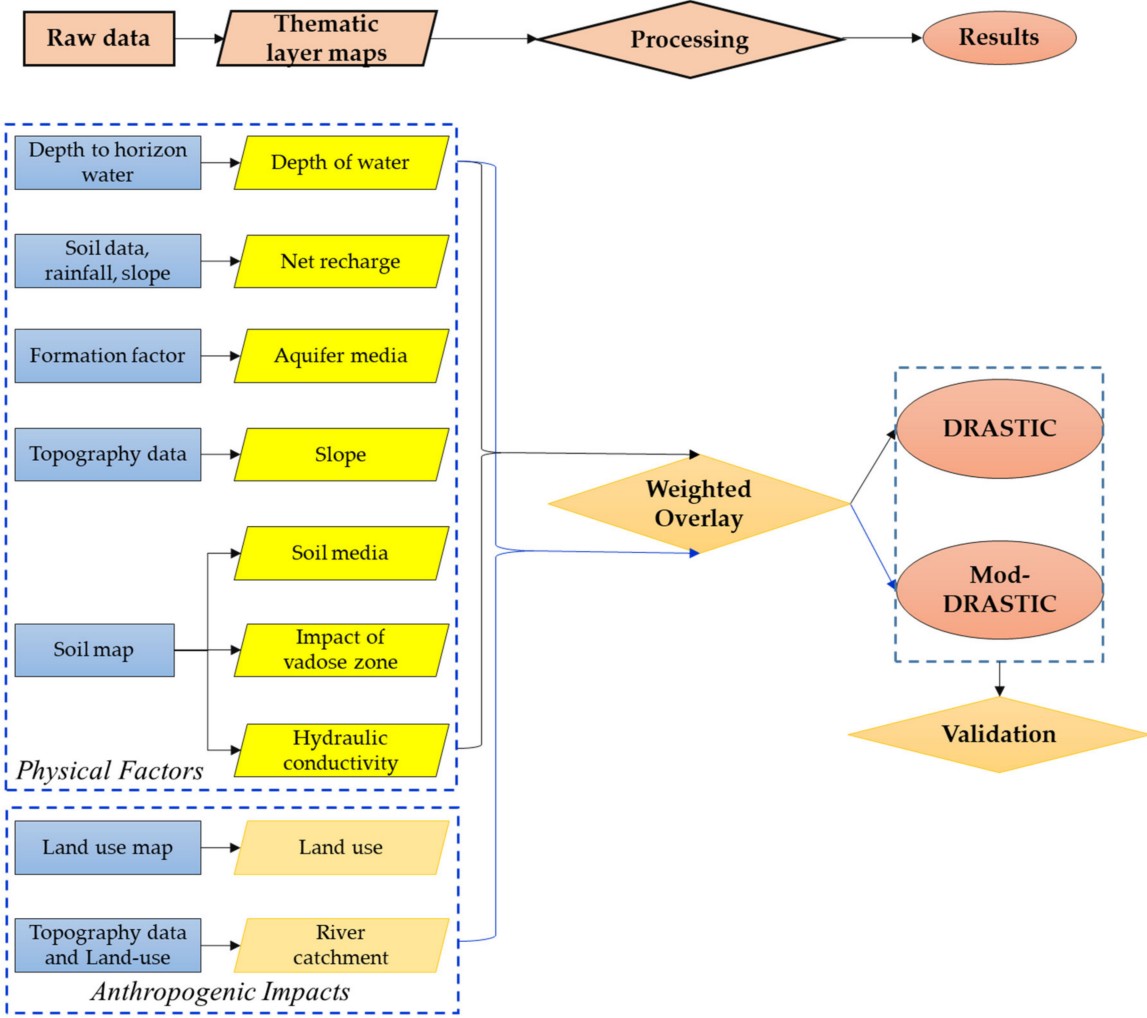

**Figure 2.** Flowchart of the research method used to generate soil water vulnerability maps using the generic DRASTIC and Mod-DRASTIC models. In addition to seven factors that were used in the DRASTIC model (shown in yellow), the Mod-DRASTIC model included two anthropogenic factors (shown in orange).

ArcGIS 10.8 was employed to create the input datasets and perform the DRASTIC model. Afterward, the Kriging interpolation, a tool that is integrated into ArcGIS, was applied to interpolate the depth of water (D) and aquifer (A) layers from available data points. The slope calculation tool in ArcGIS 10.8 was used to derive a raster layer of terrain slope from the Digital Elevation Model (DEM).

### 2.4.1. Depth to the Water Table

The depth of the water (D) level was assessed using a network of 150 hand-auger with a depth of 100 cm and a spatial grid of 5 km. In this study, the depth to water table, ranging from 10 to 100 cm below the ground surface, was measured using a hand ruler. The field survey water table data that were in a form of point data type were then interpolated to the continuous surface raster layer using the geostatistical interpolation model in ArcGIS. Considering that the soil water potential protection is significantly increased with the depth to water table because deeper water depths result from longer transmission times [3], the maximum weight of the five scores was assigned to the depth to water table [37]. This indicates that the depth to water table is one of the most important factors among nine parameters in the Mod-DRASTIC model. Regarding the rating for different levels of the depth to water table, a standard scoring scale ranged from 1 to 10 scores was used (Table 3). This standardized scale was widely used in previous studies [17,18,32], indicatings that the deeper the water table, the smaller the rating value [38,39]. In the study area, the lowest rating (i.e., 1 point) was given to deeper water tables (e.g., the region around Cho Lach), and the maximum one (i.e., 10 points) was given to shallow water tables (e.g., around Thanh Phu) (Figure 3a). The largest area of 10–20 cm in depth (49.11%) was distributed in most of the districts in the study area, followed by the depth of 20–30 cm (26.14%).

**Table 3.** Adjusted DRASTIC factor default ratings ($R_i$) used to predict the soil water's vulnerability to salinization in the study area.

| Factor | Rating | Index | Land Area (km$^2$) | Area (%) |
|---|---|---|---|---|
| **Depth to Water Level (cm)** | | | | |
| 0–10 | 10 | 50 | 358.88 | 18.41 |
| 10–20 | 8 | 40 | 957.12 | 49.11 |
| 20–30 | 6 | 30 | 509.48 | 26.14 |
| 30–40 | 4 | 20 | 53.28 | 2.73 |
| 40–60 | 2 | 10 | 67.32 | 3.45 |
| >60 | 1 | 5 | 2.91 | 0.15 |
| Net Recharge (Amount of water) | | | | |
| Zone 1 | 1 | 4 | 582.81 | 29.90 |
| Zone 2 | 2 | 8 | 1273.72 | 65.35 |
| Zone 3 | 3 | 12 | 92.47 | 4.74 |
| Aquifer media (Formation factor) | | | | |
| 0–2 | 1 | 3 | 527.32 | 27.06 |
| 2–4 | 2 | 6 | 775.96 | 39.81 |
| >4 | 3 | 9 | 645.72 | 33.13 |
| Soil media | | | | |
| Silty clay (high salinity) | 1 | 2 | 329.44 | 16.90 |
| Silty clay (moderate and low salinity) | 2 | 4 | 278.94 | 14.31 |
| Silty clay (moderate salinity, acid sulfate) | 3 | 6 | 75.70 | 3.88 |
| Silty clay (acid sulfate) | 4 | 8 | 76.30 | 3.91 |
| Silty clay (rowed—acid sulfate) | 5 | 10 | 277.20 | 14.22 |
| Silty clay (alluvial with yellowish red mottles) | 6 | 12 | 258.29 | 13.25 |
| Silty clay loam (rowed—alluvial) | 7 | 14 | 552.97 | 28.37 |
| Sandy clay loam (sandy) | 8 | 16 | 100.15 | 5.14 |
| Topography (% Slope) | | | | |
| 0–0.02 | 3 | 6 | 598.78 | 30.72 |
| 0.02–0.1 | 2 | 4 | 1017.09 | 52.19 |
| >0.1 | 1 | 2 | 333.12 | 17.09 |
| Impact of vadose zone | | | | 100.00 |
| Sand | 3 | 15 | 100.15 | 5.14 |
| Silt | 2 | 10 | 1365.67 | 70.07 |
| Clay | 1 | 5 | 483.17 | 24.79 |
| Hydraulic Conductivity (m/day) | | | | |
| 0.2–0.5 (sand) | 3 | 12 | 100.15 | 5.14 |
| 0.2–0.002 (silt) | 2 | 8 | 1365.67 | 70.07 |

**Table 3.** *Cont.*

| Factor | Rating | Index | Land Area (km$^2$) | Area (%) |
|---|---|---|---|---|
| **Depth to Water Level (cm)** | | | | |
| <0.002 (clay) | 1 | 4 | 483.17 | 24.79 |
| River catchment | | | | |
| Ham Luong (0.17) | 1 | 5 | 1082.54 | 55.54 |
| Dai (0.19) | 2 | 10 | 544.74 | 27.95 |
| Co Chien (0.23) | 3 | 15 | 321.72 | 16.51 |
| Land use | | | | |
| Other Land Types | 1 | 5 | 41.54 | 2.13 |
| Forest | 2 | 10 | 74.65 | 3.83 |
| Crop Land | 3 | 15 | 1528.06 | 78.40 |
| Aquaculture | 4 | 20 | 282.05 | 14.47 |
| Salt Production | 5 | 25 | 22.70 | 1.16 |

2.4.2. The Net Recharge (R)

The recharge water is defined as 'a significant vehicle for percolating and transporting saltwater from the vadose zone to the saturated zone' [40]. In this study, the Piscopo's (2001) method [41], as presented in Equation (2), was applied to calculate net recharge:

$$\text{Amount of water} = \text{soil permeability} + \text{rainfall} + \text{slope present} \tag{2}$$

where soil permeability was determined based on the soil map of Ben Tre at a scale of 1:50,000, in which the grain size distribution was classified as sand, silt, and clay (USDA system); the rainfall was from the annual average rainfall recorded by the Centre for Meteorology and Hydrology in Ben Tre; and the slope was taken from the Ben Tre topography map. Generally, the greater the recharge, the greater the probability of the salinity content reaching the water table [14,18]. However, in the study area, soil water salinization is reduced if the net recharge increases because the seawater intrusion mainly depends on the river or canal network [9,30]. As such, the net recharge is categorized into three zones and assigned a rating of 1, 2, or 3 (Figure 3b and Table 3). The highest rating of 3 was allocated to the highest net recharge zones that enable the penetration of saltwater from the surface to the water table. In contrast, the lowest rating of 1 was assigned for areas that prevent downward and lateral penetration of saltwater.

2.4.3. Aquifer Media (A)

The aquifer media presents the aquifer's salinity attenuation capacity, which depends on the quantity and types of the grains [17,36]. Because the formation factor (F) leans on the porosity (Φ) (i.e., the grain size of aquifer media), the greater the (F), the higher the salinization possibility [4,42]. In this study, the aquifer media were obtained indirectly using the formation factor in Archie's empirical formula [43–45], as in Equation (3), and the raster map was prepared via the interpolation method.

$$F_i = a\Phi^m = EC_{bulki} / \sigma_{wi} \tag{3}$$

where $EC_{bulki}$ represents the electrical conductivity of the second geo-electrical layer from the layered resistivity model; a is the constant coefficient; Φ is the porosity; m represents the cementation coefficients; $\sigma_{wi}$ is soil-water electrical conductivity; and F is the formation factor, which depends on a, Φ, and m at each of the hand-auger holes.

The aquifer media that include three classes (0–2, 2–4, and >4) were characterized from the formation factor map using Equation (3) and assigned a rating of 1, 2, or 3, respectively (Figure 3c and Table 3). In the study area, a formation factor of 4 or above accounted for 33.13% of the total land area, followed by a value of 2 to 4 with 39.81%.

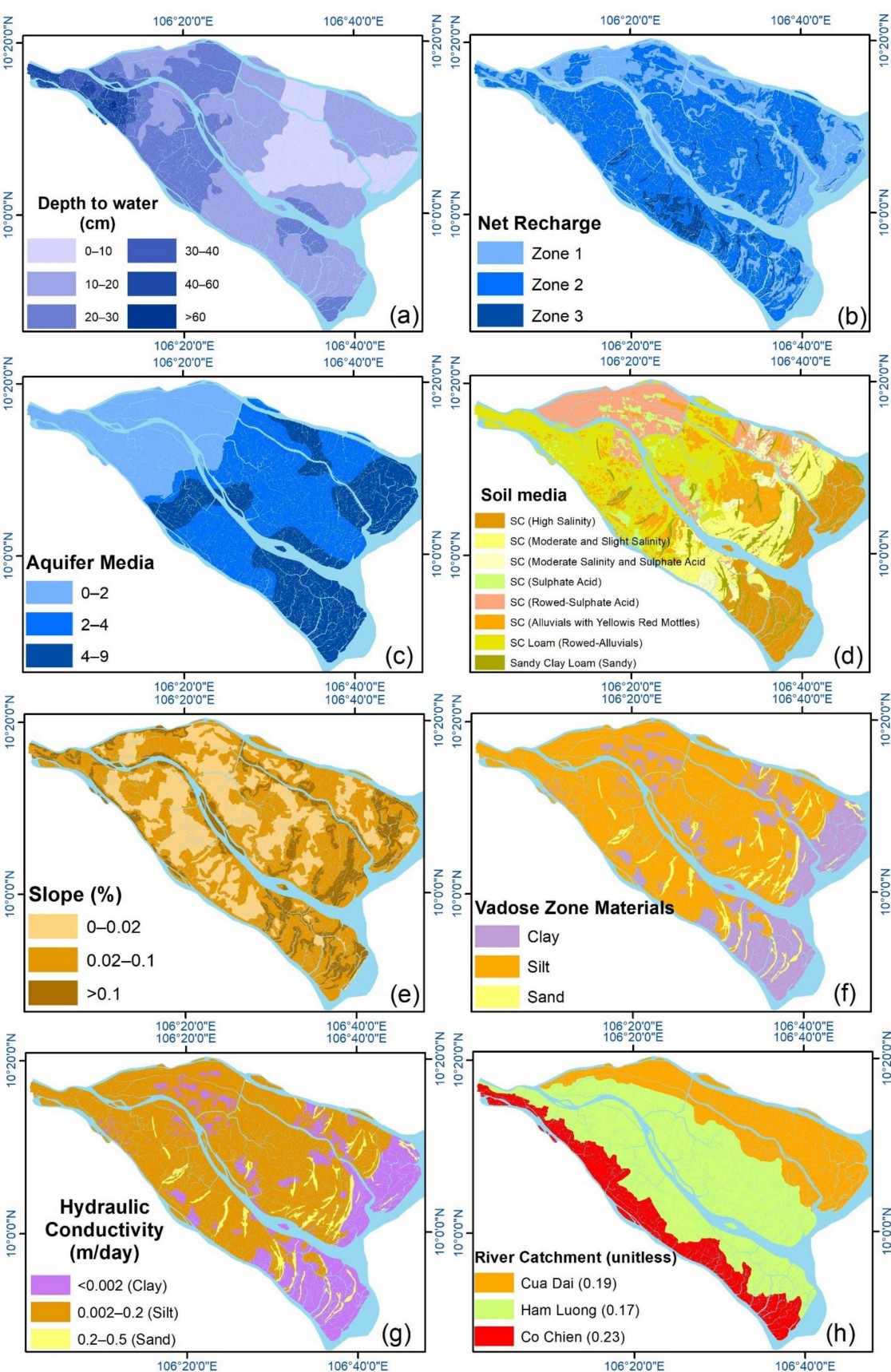

**Figure 3.** *Cont.*

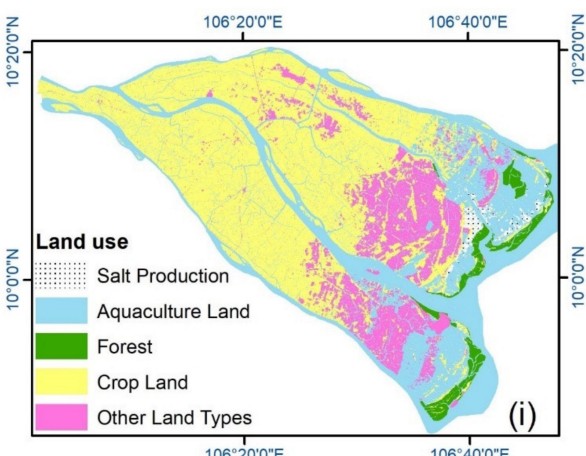

**Figure 3.** Thematic maps for DRASTIC and Mod-DRASTIC models (i.e., (**a**) depth of water, (**b**) net recharge, (**c**) aquifer media, (**d**) soil media, (**e**) slope, (**f**) vadose zone materials, (**g**) hydraulic conductivity, (**h**) river catchment, and (**i**) land use).

### 2.4.4. Soil Media

The soil in the uppermost part of the unsaturated zone affects the salinization ability to move within the soil down to the water table [3]. The soil medium specifies the levels of groundwater vulnerability because it has a direct impact on the quantity of groundwater recharge and the capability of saline water to penetrate through the vadose zone [46]. Considering that the water-holding capacity varies with different soil types, soil media can significantly influence the transmission time for the salinity [47]. The existence of fine-grain materials in the soil media such as silt or clay can reduce the permeability and slowdown salinity transfer through physicochemical processes [48]. The soil map showed that silty clay and sandy clay were the dominant soil types in the study area. As the amount of clay increases, the vulnerability to salinization decreases [17]. In this context, the Thanh Phu, Binh Dai, and Ba Tri districts face the highest risk of salinization, while the inland districts have relatively lower susceptibility to salinity. The rating and distribution of soil media are shown in Table 3 and Figure 3d, respectively.

### 2.4.5. Topography

Given that flat areas reserve water for a long period and have higher infiltration capacity, surface water recharge occurs, and this process is a significant potential for salinity migration. In contrast, steep-lands are prone to surface runoff, and the least infiltration occurs, making them less vulnerable to salinization [30]. Therefore, a smaller slope indicates higher salinization. In the study area, gentle slopes (i.e., 0–0.02% and 0.02–0.1%) accounted for 30.72% and 52.19%, respectively, and thus were assigned DRASTIC scores of 3 and 2, as presented in Table 3 and Figure 3e.

### 2.4.6. Impact of Vadose Zone

The vadose zone is the area between the saturated soil layer and the surface soil layer [49]. Its influence on the soil water salinity potential is similar to that of soil cover, which depends on permeability and the attenuation characteristics of the media [4]. The vadose zone materials control the passage and attenuation of saline water into the soil-water-bearing layer. In this study, a rating from 1 to 3 was assigned to the relevant vadose zone prior to the major soil texture (Figure 3f and Table 3).

### 2.4.7. Hydraulic Conductivity

This factor refers to the ability of the soil water-bearing layer to transmit water and control salinity migration [47]. Herein, the hydraulic conductivity values for different soil media determined by Ritzema (2006) [50,51] were used, as presented in Table 4.

**Table 4.** Hydraulic conductivity: K-value range by soil texture.

| Texture | Hydraulic Conductivity K (m/day) |
|---|---|
| Gravelly, coarse sand | 10–50 |
| Medium sand | 1–5 |
| Sandy loam, fine sand | 1–3 |
| Loam, clay loam, clay (well-structured) | 0.5–2 |
| Very fine sandy loam | 0.2–0.5 |
| Clay loam, clay (poorly structured) | 0.002–0.2 |
| Dense clay (no cracks, pores) | <0.002 |

A soil-water-bearing layer that has a high hydraulic conductivity is more susceptible to salinization than others. Hydraulic conductivity was divided into three ranges: below 0.002, 0.002–0.2, and 0.2–0.5 m per day, where the lowest range receives a rating of 1, and the highest gets a rating of 3 (Table 3). Most of the districts in the study area were covered by the hydraulic conductivity of 0.002–0.2 m/day, accounting for 70.07% of the total land area (Figure 3g). Therefore, the research revealed a low to moderate vulnerability to soil water salinity in the study area.

2.4.8. River Catchment

The ratio of river/canal surface area to the corresponding river catchment area is one of the vital conditions for saline intrusion. Saltwater intrusion may be transmitted from the sea inland via a river or canal network and then infiltrates to soil layers [33,52]. The province is on the island in the Mekong River mouth, formed by an alluvial deposition process with a densely interlaced river/canal system accounting for a total area of 445.82 km$^2$ (18.62%) [53]. In addition, agricultural activities and the subsequent draining of fertilizers by the surface streams are typical across the study site. Hence, river catchment is one of the important factors for modeling soil water susceptibility to salinization in the study area, because the impact of surface stream density on saltwater intrusion and ground water quality may highly differ across river catchments. A map of the three rivers' catchments was constructed with three ratios—0.17, 0.19, and 0.23 (Figure 3h)—and the rating of each is presented in Table 3.

2.4.9. Land Use Types

The significant impact of the land use type on the groundwater's susceptibility to salinization has been observed in different regions in the world [18]. The patterns of various land use enable the control of salinity content variation and its infiltration to a soil-water-bearing layer. The agricultural land, for example, is most responsible for the decrease in soil water quality in the study area. The provincial land use map was categorized into five classes: aquaculture, crop land, forest, salt production, and other land types. It showed that crop land is the dominant land use type in the study site, with an area of 1528.06 km$^2$ (78.40%), followed by aquaculture (282.05 km$^2$; 14.47%) and forest (74.65 km$^2$; 3.83%). Although the area of salt production was only 22.70 km$^2$, which accounted for 1.16%, it was a major factor in land use types creating salinization risk (Figure 3i). Therefore, the highest rating was assigned to the salt production category, and the rest of the land use types were allocated reasonable or minimum ratings (Table 3). The highest rating was assigned to the salt production category. The rest of the land use types were given minimum ratings, with an overall weight of 5 for the land use factor.

2.5. Validation

Validation is essential to produce reliable results, and therefore this process ensures that models' outcomes can be used as the scientific basis for practical applications [32]. The vulnerability map is assessed through the total dissolved solids (TDS) concentration, which is the sum of all the substances, organic and inorganic, and dissolved in water. In fact, electrical conductivity (EC) that uses the hand-auger for sample collection has been used

for decades to measure salinity [35,54,55]. However, all natural water resources contain some dissolved solids (salinity) due to contact with soil, rocks, and other materials (e.g., sediments) [56]. Therefore, using the TDS concentration to reflect the salinity content is more accurate when considering the conditions of other materials in accordance with the specific surroundings [57,58]. Many studies have also proved the significant relationship between EC measurement and TDS in water [56,59,60]. In this study, 150 samples of soil water TDS concentrations (Figure 1) were used for calibration as the main content that was introduced into the soil water environment from hydrogeological settings and anthropogenic activities. A regression analysis between TDS concentration and both models (i.e., classic and modified) was implemented. A land use map was utilized to identify potential sources of salinization in the study area. Furthermore, the modified DRASTIC map's accuracy was examined by examining the level of vulnerability in each area and its relation to the land use map.

## 3. Results

### 3.1. Soil Water Vulnerability to Salinization

By using the weighted overlaying of seven or nine layers, vulnerability maps were obtained from the classic and adjusted DRASTIC models. The final vulnerability indices varied from 36 to 128 and from 55 to 163 for the generic and modified DRASTIC model, respectively. Applying the Natural Breaks (Jenks) classification method, the values and categories of the susceptibility indices (i.e., low, moderate, high, and very high) were classified and illustrated in Figure 4. The classic DRASTIC model with seven thematic factors was categorized into four susceptibility classes (i.e., low (36–72), moderate (72–85), high (85–99), and very high (99–128)) (Figure 4a), while the modified DRASTIC with nine factors contained four classes, namely, low (55–90), moderate (90–104), high (104–118), and very high (118–163) (Figure 4b). The standard classification of the indices was found in the research published by Aller et al. (1987) [37]. Both models revealed that the province mainly faces moderate and high susceptibilities to salinization, accounting for over 60% and 70% for the generic and adjusted DRASTIC, respectively. Moreover, the vulnerable indices decreased from the coastal area to inland districts. These values demonstrated that there was no very high susceptibility around inland districts (i.e., Cho Lach, Chau Thanh, and Ben Tre City), while in the coastal districts (e.g., Ba Tri and Binh Dai), the very high level accounted for over 20% (Table 5). Additionally, both models showed a low-vulnerability zone in Cho Lach, Chau Thanh, and Ben Tre City, with a percentage of 30% or above, while the high and very high vulnerability zones covered over 40% of the total land area in coastal districts (i.e., Ba Tri, Thanh Phu, and Binh Dai). This is because, around coastal areas, the shallow depth of the soil water and the high soil media rating result in high vulnerability to salinization, while a low vulnerability index is caused by deeper soil water and low rating values for the factors of the recharge rate, formation factor, soil media, slope, vadose zone, and hydraulic conductivity around the inland districts.

Results of the classic DRASTIC showed that the very high vulnerability zone covered 44% of the land area in Ba Tri, followed by 34% in Giong Trom. In comparison, the Mod-DRASTIC model explained 16% and 12% of the presence of the same category in a similar area. In fact, the salinity around the Ba Tri, Giong Trom, and Binh Dai districts has been controlled since the Ba Lai dams were established after 2000 and the salt-prevented dykes were built in 2011 [30]. Evidence was also presented by Hoa et al. (2019) and Tran et al. (2019, 2020) [8,9,61]. Therefore, from a comparative analysis of the two models (i.e., generic and modified DRASTIC), it demonstrates that in the study area, the Mod-DRASTIC model is better than the generic DRASTIC model for mapping the soil water susceptibility to salinization.

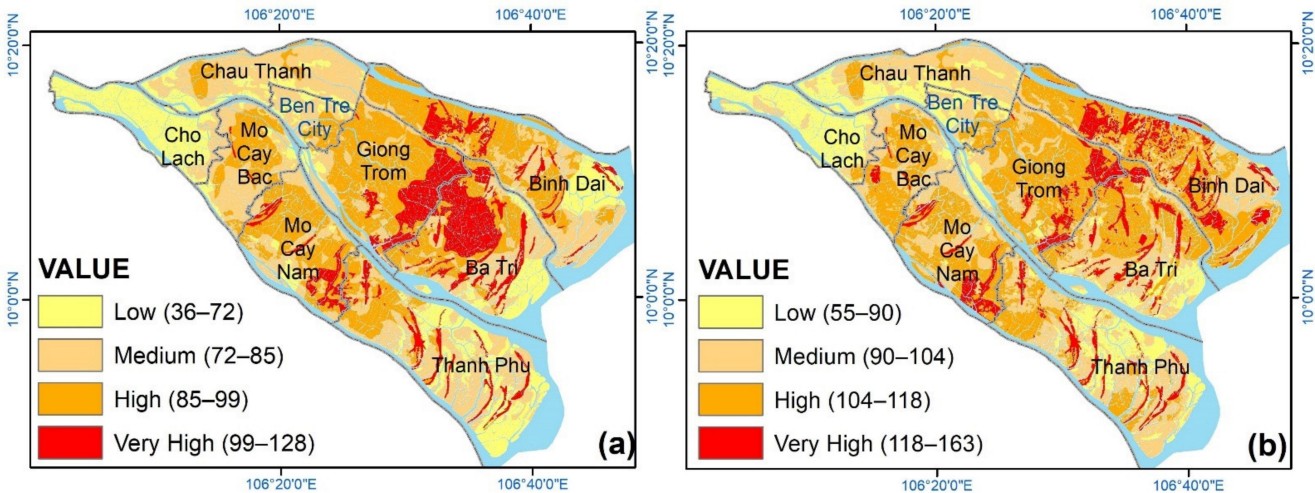

**Figure 4.** The levels of saltwater intrusion risk determined using (**a**) generic DRASTIC and (**b**) Mod-DRASTIC models.

**Table 5.** The vulnerability level area by district in the study area (%).

| District | Low | | Moderate | | High | | Very High | |
|---|---|---|---|---|---|---|---|---|
| | DRASTIC | Mod-DRASTIC | DRASTIC | Mod-DRASTIC | DRASTIC | Mod-DRASTIC | DRASTIC | Mod-DRASTIC |
| Ba Tri | 11 | 9 | 23 | 29 | 22 | 46 | 44 | 16 |
| Ben Tre city | 15 | 35 | 54 | 48 | 31 | 17 | - | - |
| Binh Dai | 14 | 1 | 31 | 26 | 43 | 52 | 12 | 22 |
| Chau Thanh | 20 | 32 | 71 | 58 | 9 | 10 | - | - |
| Cho Lach | 61 | 58 | 18 | 20 | 21 | 22 | - | - |
| Giong Trom | 2 | 3 | 12 | 21 | 52 | 63 | 34 | 12 |
| Mo Cay Bac | 6 | 3 | 54 | 44 | 39 | 50 | 1 | 2 |
| Mo Cay Nam | 3 | 1 | 37 | 26 | 49 | 64 | 11 | 9 |
| Thanh Phu | 27 | 20 | 40 | 43 | 17 | 29 | 16 | 8 |
| Total | 16 | 14 | 35 | 33 | 32 | 43 | 17 | 10 |

### 3.2. Validation with Total Dissolved Solids (TDS) Concentration

The spatial distribution of the TDS concentrations of 150 soil water samples is illustrated in Figure 1. A comparison between the field survey TDS of soil water content and the vulnerable indices maps was applied to evaluate the accuracy of the model outcomes (Figure 5). The assessment results showed that the TDS concentration varies between 90 and 34,690 mg/L (Table 6). The TDS concentration was recorded in the moderate-risk zone at 90–30,160 mg/L, while the content of 130–14,280 mg/L was found in the low-risk zone. Around the high- and very-high-risk zones, the TDS contents were 100–34,960 mg/L and 960–16,770 mg/L, respectively. Compared to the maximum permitted value of 2000 mg/L, following the FAO standard [56,62], the mean values of the TDS concentration in the moderate, high, and very high zones were at least twice as high as they should be. This demonstrated that the freshwater in the soil-water-bearing layer was easily threatened by seawater impacts around areas with moderate susceptibility (e.g., coastal districts). However, a mean value of 1844.38 mg/L was recorded in the low-risk zone, mainly Cho Lach, Chau Thanh, and Ben Tre City.

The statistical analysis of both models and the TDS concentration was employed to explain the importance of the hypothesis. With the obtained correlation coefficients of 0.69 and 0.76 ($p < 0.05$) for the classic DRASTIC and the Mod-DRASTIC indices, it is concluded that the Mod-DRASTIC model is better than the classic DRASTIC model at estimating the soil water susceptibility to salinization in the province.

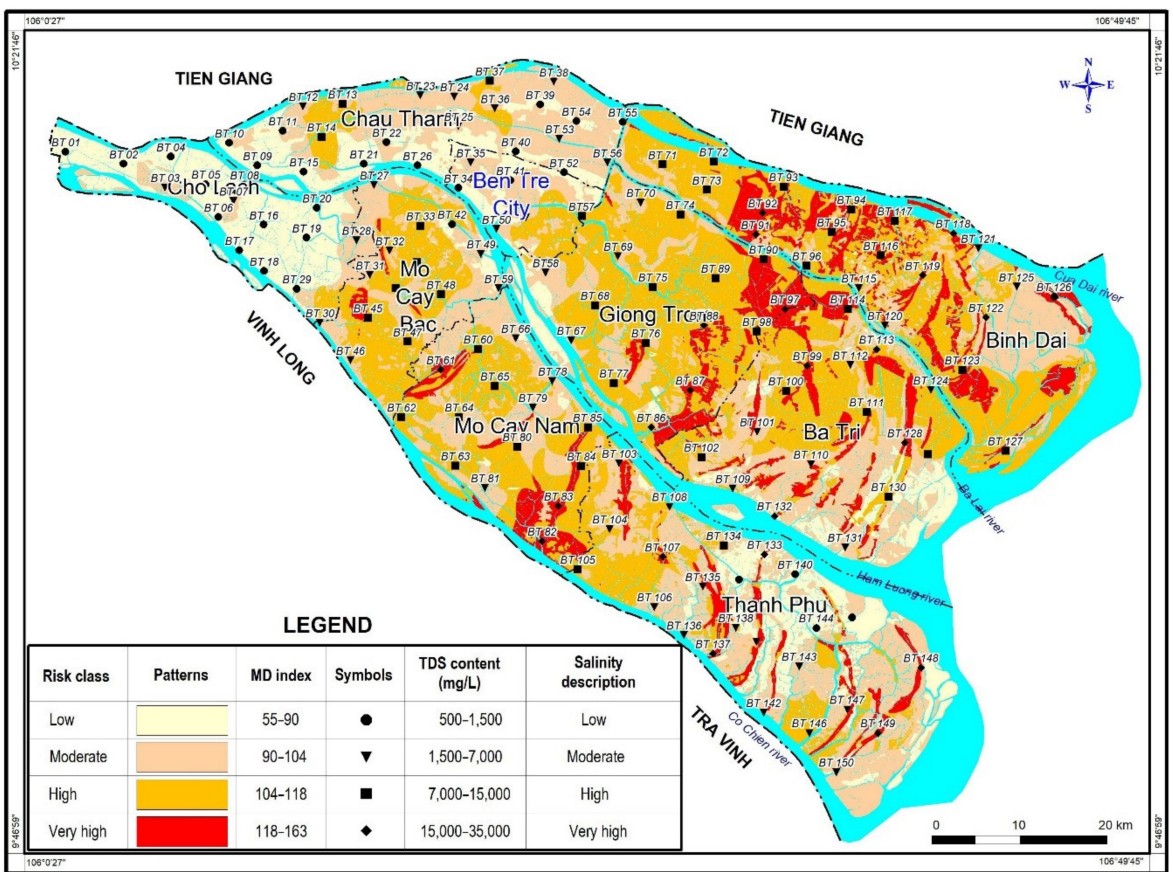

**Figure 5.** The spatial distribution of TDS content and Mod-DRASTIC vulnerability categories in the study area.

**Table 6.** Analyzing TDS concentrations on the salinization risk map in the study area.

| TDS (mg/L)    Risk | Low | Moderate | High | Very High |
|---|---|---|---|---|
| Max | 14,280 | 30,160 | 34,960 | 16,770 |
| Min | 130 | 90 | 110 | 960 |
| Mean | 1844.38 | 4841.31 | 4188.0 | 5965.24 |

## 4. Discussion

The variations in saline content in the soil-water-bearing layer depend on geo-hydrological settings and anthropogenic activities in a specific study area. Even though the classic DRAS-TIC model with seven factors has commonly been used for groundwater vulnerability assessment, several studies have pointed out that modeling this without considering the impacts of human activities is a significant limitation [14,18,32,33]. To overcome this gap, the input of the classic DRASTIC model has been adjusted by adding or eliminating some generic factors, following the local conditions [32,33]. In this study, the classic DRASTIC model was adapted to create the Mod-DRASTIC model by adding two factors (i.e., river catchment and land use), based on the important role of these local features in the study area. Seawater intrusion naturally occurs in the dry season with densely populated river and canal networks as it transmits quickly to inland.

Besides, the relationship between soil water vulnerability to salinization and the land use category in the study area was discovered in this study (Table 7 and Figure 6). Both models revealed that the areas at moderate and high risk of salinization were aquaculture and crop land (over 60%). This demonstrated that most of the areas with high and very high susceptibility to salinity were used for salt production, while forest showed low to moderate vulnerability. In aquaculture land, the different forms of breeding can lead

to variations in the salinity category. For instance, high and very high vulnerability to salinization was seen in shrimp farming areas, while low and moderate risk was found in fish farming areas. Obviously, the presence of the river catchment of Ham Luong, crop land, and other land use categories were able to reduce the salinization risk, while the salt production land use category might enhance the vulnerability to saltwater intrusion in the soil-water-bearing layer.

**Table 7.** The percentage of area at salinization risk by land use types in the study area (%).

| Land Use | Low | | Moderate | | High | | Very High | |
|---|---|---|---|---|---|---|---|---|
| | DRASTIC | Mod-DRASTIC | DRASTIC | Mod-DRASTIC | DRASTIC | Mod-DRASTIC | DRASTIC | Mod-DRASTIC |
| Aquaculture | 18 | 10 | 30 | 29 | 39 | 48 | 13 | 13 |
| Crop Land | 11 | 15 | 24 | 24 | 29 | 52 | 36 | 9 |
| Forest | 60 | 25 | 22 | 40 | 1 | 25 | 18 | 10 |
| Salt Production | 29 | 3 | 44 | 13 | 22 | 69 | 5 | 15 |
| Other Land Types | 4 | 6 | 17 | 22 | 22 | 65 | 58 | 7 |

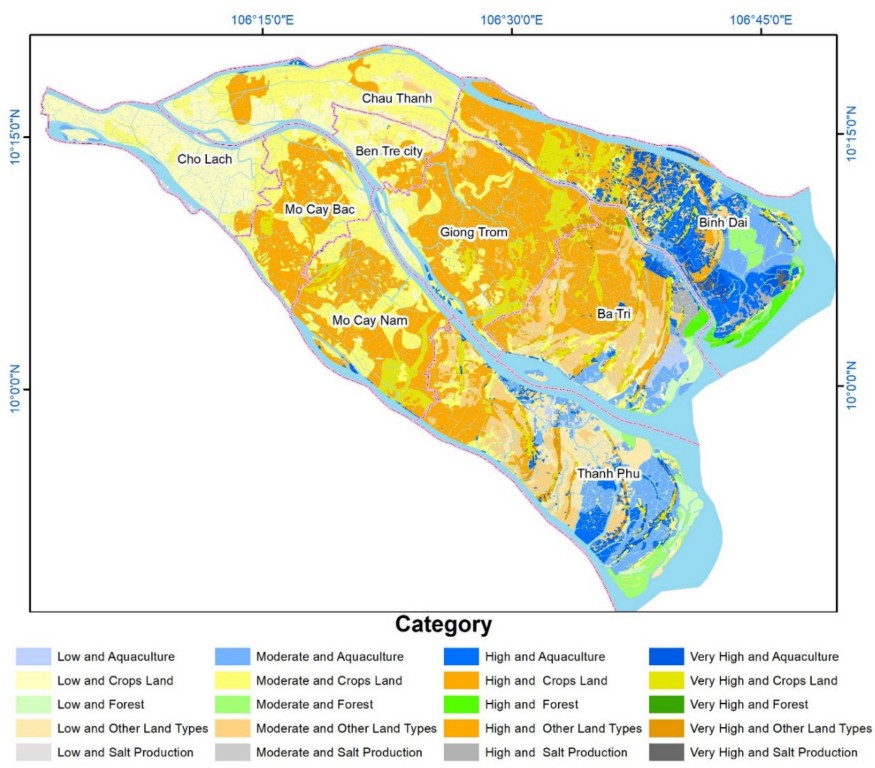

**Figure 6.** The Mod-DRASTIC vulnerability categories and land use types.

From the results of the Mod-DRASTIC, 69% and 15% of the total land in salt production was discovered to be at high or very high risk of salinization. By contrast, only 22% and 5% of the categories were found in the classic model (Table 6). Therefore, with a combination of river catchment and land use types, known as anthropogenic activities, an improvement of the map in terms of the soil water susceptibility to salinization can be explored in comparison to the local parameters.

## 5. Conclusions

The risk of soil water salinization in the Ben Tre province was evaluated using the generic DRASTIC and Mod-DRASTIC indices. The generic DRASTIC model was modified by incorporating the river catchment and land use layers to analyze the susceptibility to salinization of water in the soil-water-bearing layer in the Ben Tre province, located in

the Mekong River Delta. Nine factors were considered for quantifying the vulnerability index, including seven thematic elements of the classic DRASTIC model (i.e., depth of water, net recharge, aquifer media, soil media, topography, vadose zone, and hydraulic conductivity) and two additional parameters (i.e., river catchment and land use pattern), known as the anthropogenic impacts. Weights and ratings of the principal factors were assigned to produce vulnerability maps following the local conditions. Field collection and laboratory analysis of soil water TDS concentration were also used to validate the results.

The results of this study revealed that the coastal districts (i.e., Ba Tri, Thanh Phu, and Binh Dai) were vulnerable areas with high to very high salinity, while around the inland districts (i.e., Cho Lach, Chau Thanh, and Ben Tre City), low to moderate vulnerability to the salinization of the water in the soil layer was recorded. The combination of the river catchment, land use types, and thematic factors of the classic DRASTIC model allowed for the validation of the Mod-DRASTIC model in conjunction with the TDS concentration ($R^2 = 0.76$). Furthermore, the potential of seawater intrusion was assessed for the land use map of the Ben Tre province, and we observed that over 60% of the total aquaculture and crop land area faced a moderate to high salinization risk, whereas 69% and 15% of the total land for salt production was discovered to have high or very high vulnerability to salinization. Therefore, adding the two anthropogenic factors associated with LULC and river catchment in the study area enabled us to model the vulnerability to salinization better. Hence, we can conclude that the additional factors of LU and RC should be involved in the assessment of vulnerability in the MRD.

The methods applied in this study will serve as a valuable reference for future work in land use conversation and hazard analysis, as they can be used worldwide in any other coastal areas, considering any time frames and any biomes. This approach can provide a comprehensive understanding of soil-water-bearing-layer vulnerability to salinization. This will help policymakers and planners to develop appropriate policies in coastal regions to adapt to and reduce the effects of global warming in connection to a water level reduction upstream. In particular, the local government can apply the results to transform land use to mitigate the vulnerability to salinization.

**Author Contributions:** T.N.L., D.X.T., and T.V.T. conceived the idea of the study. T.V.L., T.H.L., D.Q.N., and T.V.D. conducted the field work and laboratory analysis. T.N.L., D.X.T., T.V.T., and S.G. processed the data and analyzed the results. All authors contributed to the writing the manuscript. All authors have read and agreed to the published version of the manuscript.

**Funding:** This research was funded by the Department of Science and Technology of Ben Tre province under the project entitled "Determining causes, forecasting saline intrusion into soil and soil-water in Ben Tre province in the context of climate change and sea level rise. Proposal of appropriate adaptation measurements", following the Decision of 304/QĐ-UBND, issued on 17/02/2017 by the People's Committee in Ben Tre province.

**Institutional Review Board Statement:** Not applicable.

**Informed Consent Statement:** Not applicable.

**Data Availability Statement:** Not applicable.

**Acknowledgments:** The authors would like to thank the Ho Chi Minh City Institute of Resources Geography, Vietnamese Academy of Science and Technology, and Department of Science and Technology of the Ben Tre province for supporting software and data.

**Conflicts of Interest:** The authors declare no conflict of interest. The funders had no role in the design of the study; in the collection, analyses, or interpretation of data; in the writing of the manuscript and even in the decision to publish the results.

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
