# Peer review of "Estimating Soil Water Susceptibility to Salinization in the Mekong River Delta Using a Modified DRASTIC Model"

_water, doi:10.3390/w13121636_

Round 1

Reviewer 1 Report

The paper deals with a very emergent topics and the approach can be interesting but I definitely can't understand which is the integration authors proposed for  DRASTIC, whose application needs some more detailed explanations. The last proposed factors needs to be better explained to let readers and reviewers its real innovation and effectiveness of it.

Author Response

Dear Reviewer,

Please find enclosed the response to the reviewers’ comments as well as the new manuscript version for our paper (ID: water-1183130) “Estimating Soil Water Susceptibility to Salinization in the Mekong River Delta Using a Modified DRASTIC Model”, to be considered for publication in Water.

We have done our best to address the reviewers’ suggestions and concerns in the answering letter as well as in the revised manuscript. All the changes in the manuscript have been highlighted using Track changes mode and making in red.

The entire manuscript has been revised extensively, and we hope that it is now commensurate with the high standard of the Water.

If you need any further information or clarification, please do not hesitate to contact us.

Looking forward to hearing from you.

Sincerely,

Author (s)

Reviewer 2 Report

In the manuscript, the authors present an estimation of soil water sensitivity to contamination (salinization)  using the modification of the generally accepted DRASTIC method. Modification of the method refers to an increase in the number of factors involved in calculating the vulnerability index.
For the study site, this modified version provided better results in comparison to the generic method.
Data collection included fieldwork forming the spatial grid of 150 sampling sites over 5 km, providing a significant number of samples to perform the analysis.
Generating soil-water vulnerability maps is well presented by flow charts.
Results would be of interest for the planners and decision-makers in the regions with similar hydrological and hydrogeological conditions to ones presented in the paper i.e. river deltas with a significant risk of salinization.
However, the manuscript is occasionally difficult to read due to grammatical errors and clumsy expressions which is why I recommend proofreading by a native speaker.

Author Response

(The authors gave the same response as above.)
